

# consensusDE: an R package for assessing consensus of multiple RNA-seq algorithms with RUV correction

Ashley J. Waardenberg[1] and Matthew A. Field[1,2]

[1] Australian Institute for Tropical Health and Medicine, Centre for Tropical Bioinformatics and Molecular Biology, Centre for Molecular Therapeutics, James Cook University, Smithfield, Australia
[2] John Curtin School of Medical Research, Australian National University, Canberra, Australia

## ABSTRACT

Extensive evaluation of RNA-seq methods have demonstrated that no single algorithm consistently outperforms all others. Removal of unwanted variation (RUV) has also been proposed as a method for stabilizing differential expression (DE) results. Despite this, it remains a challenge to run multiple RNA-seq algorithms to identify significant differences common to multiple algorithms, whilst also integrating and assessing the impact of RUV into all algorithms. consensusDE was developed to automate the process of identifying significant DE by combining the results from multiple algorithms with minimal user input and with the option to automatically integrate RUV. consensusDE only requires a table describing the sample groups, a directory containing BAM files or preprocessed count tables and an optional transcript database for annotation. It supports merging of technical replicates, paired analyses and outputs a compendium of plots to guide the user in subsequent analyses. Herein, we assess the ability of RUV to improve DE stability when combined with multiple algorithms and between algorithms, through application to real and simulated data. We find that, although RUV increased fold change stability between algorithms, it demonstrated improved FDR in a setting of low replication for the intersect, the effect was algorithm specific and diminished with increased replication, reinforcing increased replication for recovery of true DE genes. We finish by offering some rules and considerations for the application of RUV in a consensus-based setting. consensusDE is freely available, implemented in R and available as a Bioconductor package, under the GPL-3 license, along with a comprehensive vignette describing functionality: http://bioconductor.org/packages/consensusDE/.

Corresponding author
Ashley J. Waardenberg,
a.waardenberg@gmail.com

## INTRODUCTION

Differential gene expression (DE) analysis aims to identify transcripts or features that are expressed differently between conditions. For the detection of significant DE genes, a number of Bioconductor/R (*Gentleman et al., 2004*) packages have been developed that implement different statistical models for assessing DE significance. Reviews of RNA-seq DE method performance have highlighted large sensitivity and specificity differences

between methods (*Rapaport et al., 2013*; *Seyednasrollah, Laiho & Elo, 2015*). The confident selection of genes that are truly DE is especially important when trying to define reliable markers, e.g., as prognostics (*Seyednasrollah, Laiho & Elo, 2015*).

Currently, there is no gold standard approach for the analysis of RNA-seq data. Whilst there remains no gold standard it is important that attention is restricted to the best performing methods, to minimise negative results that are the consequence of technical limitations. Benchmark analyses of existing algorithms for detection of DE are important for the community to assess performance of different methods and existing analyses (*Costa-Silva, Domingues & Lopes, 2017*; *Rapaport et al., 2013*) have popularised leading algorithms, such as edgeR (*McCarthy, Chen & Smyth, 2012*; *Robinson, McCarthy & Smyth, 2010*), DESeq2 (*Love, Huber & Anders, 2014*) and limma/voom (*Ritchie et al., 2015*). Lin and Pang et al. recently proposed selecting a "best" method based on ranking DE stability against permutation (*Lin & Pang, 2019*). However, they found that no single DE method was stable in all cases, with permutation or bootstrap strategies also being limited by replicate number and computational demands. Another technique often used to assess performance of DE methods is 'False DE', where genes not expected to exhibit significant DE are examined (*Rapaport et al., 2013*; *Seyednasrollah, Laiho & Elo, 2015*). In negative control 'False DE' experiments it was found that DE genes generally did not overlap and were specific to individual algorithms (*Soneson & Delorenzi, 2013*). A comparison of 11 methods found that uniquely identified DE genes are often attributed to low fold changes (*Soneson & Delorenzi, 2013*), but that methods largely (with some exceptions) ranked genes similarly (*Soneson & Delorenzi, 2013*). These findings support a "combined" or consensus-based approach or at the least a need to compare and ideally benchmark results with known truth between different methods.

Removal of unwanted sources of variation (as implemented in RUVseq), is another approach that has recently been proposed to improve DE accuracy (*Risso et al., 2014*). RUV aims to improve normalization, by obtaining factors that are assumed to describe unwanted variation, and subsequently including these factors in models used for DE analysis (*Risso et al., 2014*). Three RUV methods have been proposed, RUVr, RUVg and RUVs that utilise residuals, negative control genes or technical replicates, respectively to estimate unknown factors of random variation, given a number of unknown parameters, k, that are subsequently incorporated into the DE model for estimation of transcript counts. RUV has been demonstrated to stabilize fold change, improving DE and separation of biological samples. Whether RUV generalizes across multiple algorithms and improves modelling in a "combined" or consensus-based setting has not been addressed to the best of our knowledge.

Implementing any consensus-based approach is challenging and requires combining individual algorithms that typically require different input parameters, use different method names, and generate different outputs, thus requiring the user to learn specific steps required for each package. Furthermore, correction methods, such as RUV (*Risso et al., 2014*), require users to learn additional steps for model integration. Although a number of tools have been developed for combining RNA-seq algorithms, some do not compare results from different algorithms (*Delhomme et al., 2012*; *Varet et al., 2016*), lack automation

ability outside of a web-based setting (*Jimenez-Jacinto, Sanchez-Flores & Vega-Alvarado, 2019*), are not maintained in a central repository, require additional command line knowledge for installation (*Costa-Silva, Domingues & Lopes, 2017*), implement predecessor algorithms such as DESeq (*Guo et al., 2014*; *Moulos & Hatzis, 2015*) instead of DESeq2 (*Love, Huber & Anders, 2014*) and importantly none support RUV integration.

Integration of results from different RNA-seq algorithms would ideally allow users to easily (1) import data, (2) run RNA-seq analysis across multiple algorithms, (3) require minimal parameter input, (4) offer flexible but simple options for removal of unwanted variation (e.g., RUV integration), (5) present results together in a simple table for further analysis and finally (6) provide metrics for users to determine stability of DE calls from multiple methods. Herein, we describe consensusDE, an R/Bioconductor package, which enables the above, integrating DE results from edgeR (*Robinson, McCarthy & Smyth, 2010*), limma/voom (*Ritchie et al., 2015*) and DEseq2 (*Love, Huber & Anders, 2014*) easily and reproducibly, with the additional option of integrating RUV. Through reducing the results of multiple algorithms into a single 'consensus' table with a number of descriptive statistics, users can readily assess how consistently a gene is called DE by different methods and select a consensus set for further analyses. We demonstrate the utility of consensusDE through application to real and simulated data and assess the impact of RUV for comparability or integration with multiple RNA-seq algorithms. We find that RUV improves stability of reported DE fold change and improves FDR in a setting of low replication (with largest improvements for voom). However, the application of RUV did not improve true positive rates (recall) or with increased number of replicates. We finish by offering some guidelines and considerations for application of RUV.

## MATERIALS AND METHODS

Bioinformatics analyses were performed in R version 3.5.1 (*R Core Team, 2018*) using Bioconductor (*Gentleman et al., 2004*) packages unless stated otherwise.

### consensusDE

consensusDE version 1.3.4 (BioConductor Development version) was used for all analyses. Versions of RNA-seq algorithms follow, edgeR version 3.22.5, voom (limma) version 3.36.5, DESeq2 version 1.20.0 and RUVSeq version 1.16.1. For DESeq2, cooksCutoff is disabled for comparable reporting and ranking of all $p$-values. Normalisation methods are set to the default methods for each algorithm in multi_de_pairs (norm_method ="all_defaults"). We implement RUVr, which we refer simply to RUV throughout, as described in the RUVSeq BioConductor vignette (utlising edgeR and extracting residuals), with k set to 1. RUVr removes sources of unwanted variation, by first fitting a generalised linear model (GLM) to count data and obtaining model residuals that are then incorporated into subsequent DE models. All code and parameters used in analyses are available at, https://github.com/awaardenberg/consensusDE_material and a comprehensive vignette describing functionality: http://bioconductor.org/packages/consensusDE/.

## Datasets

We apply consensusDE to both real and simulated data. Real data consisted of RNA-seq data comparing treatments to controls from human airway smooth muscle cells (*Himes et al., 2014*) while for simulated data, we obtain negative binomial distribution parameters from input read counts of real experimental data (using the simulator described here: (*Moulos & Hatzis, 2015*). For parameter estimation, mouse RNA-seq data comparing striatum of C57BL/6J and DBA/2J strains (*Bottomly et al., 2011*) was obtained from recount (*Frazee, Langmead & Leek, 2011*). Values reported are the average of 10 simulations. We simulate DE for 10,000 genes, defining the number of DE genes as 500 (5%), with equal up/down regulation and simulate DE with 3 or 5 replicates.

## Performance assessment

For assessment of RNA-seq algorithm agreement or performance we use a number of statistics. Agreement of logFC was assessed by Bland-Altman analysis (*Bland & Altman, 1986*) of the differences of logFC versus average logFC between pairs of methods, as implemented in the blandr package (version 0.5.1) (*Datta, 2017*). Bias is reported as the mean shift from zero and limits of agreement defined by 95% confidence intervals. Where $R^2$ was reported, this was the resultant goodness of fit of a linear model comparing logFC.

Concordance (or agreement) of significant DE genes called by RNA-seq algorithms was assessed using the Jaccard Similarity coefficient (JC), (Eq. (1)), representing the ratio of the size of the intersecting set *A* and *B* to the total size or union of sets *A* and *B*. Here, the size of set *A* was considered the intersecting set of all methods (voom, edgeR and DEseq2) and *B* considered either the set size of each method individually (where reported as either voom, edgeR or DEseq2) or the union of all methods (where reported as the union).

$$JC = \frac{A \cap B}{A \cup B} \tag{1}$$

Where truth was known (in the case of simulation), positive predictive value (PPV) or precision as the proportion of true positive (TP) versus false positives (FP) (Eq. (2)), sensitivity (or recall or true positive rate) as the proportion of TP versus FN (Eq. (3)) and False Discovery Rate (FDR) (Eq. (4)) were reported.

$$PPV\,(precision) = \frac{TP}{TP+FP} \tag{2}$$

$$sensitivity\,(recall) = \frac{TP}{TP+FN} \tag{3}$$

$$FDR = \frac{FP}{\sum positive\ predictions}; or\ 1-PPV \tag{4}$$

## RESULTS AND DISCUSSION

We begin by (1) describing consensusDE functionality, followed by (2) comparison to existing software and the application to (3) real and (4) simulated data for assessing performance, with and without RUV integration.
## A. Description of data

## B. consensusDE workflow

**Figure 1** **consensusDE typical workflow.** (A) consensusDE requires a table, here defined as "my_data", for example purposes, that describes the experimental design and location of files. (B) running consensusDE requires two steps, first to build a summarized object, using the buildSummarized function, to store all information and second to run analyses with all algorithms using the multi_de_pairs function. Example code for a typical analysis with consensusDE is provided for illustration.

## consensusDE functionality

consensusDE follows two simple steps for performing DE analysis with multiple RNA-seq algorithms (1) building a summarized experiment object and (2) performing DE analysis (including plotting) using the *buildSummarized* and *multi_de_pairs* functions respectively. Figure 1 provides an overview of a typical consensusDE workflow, and below we describe its core functions.

### buildSummarized

Generates a summarized experiment that contains all experimental data provided in the sample table and the read counts mapped to transcript coordinates. To build a summarized experiment object consensusDE simply requires a sample table describing the location of BAM files or pre-computed counts from the popular HTSEQ (*Anders, Pyl & Huber, 2015*), sample groupings, optional pairing or technical replicate information and transcript database information (either gtf or txdb format). Where an output directory is specified,

the user can save their compiled summarized experiment object for future analyses and/or reproducibility purposes.

### multi_de_pairs

Automatically performs DE analysis on all possible pairs of "groups" defined in a provided sample table using all available DE methods (currently edgeR, voom and DESeq2) and outputs a summary table (or merged table, described below) that merges the results of all methods into one table. Options are provided for annotations, including an option to annotate from gtf files, and users are provided the option to remove unwanted sources of variation by RUVr (*Risso et al., 2014*). RUVr, in contrast to RUVg and RUVs (which are currently not implemented in consensusDE) that rely on the assignment of negative control genes or availability of technical replicates, is readily generalisable. RUVr first fits an initial GLM fit of a supervised model to estimate unknown factors of random variation, given a number of unknown parameters (k), that are subsequently incorporated into the GLM model for each algorithm for estimation of transcript counts (referred to simply as RUV here after). The number of unknown parameters (k) is also fixed at 1. Options are provided for normalisation, permitting the same normalisation method to be applied to all RNAseq algorithms or to utilise default normalisation methods for each algorithm. Full results of each method and accessibility to model details is available (see vignette accompanying BioConductor package for details of how to access) and where output directories are provided, results and plots are automatically written to these directories, supporting batch analyses.

### Merged results

The final table of interest is described as the "merged" table (description of results provided in Table 1). The merged table contains statistics including the "p_union" representing the union *p*-value, "p_intersect" representing the intersect *p*-value and "rank_sum", being the sum of the rankings for significance of DE reported by each method. In the case of Average Expression and Log Fold Change (logFC), these represent the mean value across all methods. Standard deviation of the logFC is reported for assessment of variation of fold change between methods. Individual *p*-values (corrected for multiple hypothesis testing by default), and additional information including annotated gene symbol, gene name, kegg pathway and chromosomal coordinates are also reported when annotation is optionally selected. Thus, the merged table provides a simple summary of all methods and statistics in one location.

### Plots

Ten diagnostic plots are generated and optionally saved as pdf files; (1) mapped reads for a summary of transcript reads per sample, (2) Relative Log Expression (RLE) for quality control (QC) inspection, (3) Principle Component Analysis (PCA), (4) RUV residuals, (5) Hierarchical clustering, (6) Density distributions, (7) Boxplot, (8) Mean/Average (MA) Plot, (9) Volcano plots and 10) *p*-value histogram. For MA and volcano plots, the average logFC and averages of the average expression by each method are used. Features are coloured by significance threshold (based on the intersect *p*-value) and the size of

**Table 1  consensusDE merged table features and description.**

| Reported Feature | Meaning | Description |
|---|---|---|
| ID | Identifier | Identifier of feature used for mapping read counts against |
| AveExpr | Average Expression | Average of edgeR, DESeq2 and voom reported Average Expression |
| LogFC | Log Fold-Change, also known as a log-ratio | Average of edgeR, DESeq2 and voom logFC |
| LogFC_sd | Log Fold-Change standard deviation | Standard Deviation of LogFC reported by edgeR, DESeq2 and voom |
| edger_adj_p | edgeR $p$-value | Adjusted for multiple hypotheses using benjamini and hochberg (default) |
| deseq_adj_p | DESeq2 $p$-value | Adjusted for multiple hypotheses using benjamini and hochberg (default) |
| voom_adj_p | voom $p$-value | Adjusted for multiple hypotheses using benjamini and hochberg (default) |
| edger_rank | rank of the $p$-value reported by edgeR | smallest rank is most significant (or smallest $p$-value) reported |
| deseq_rank | rank of the $p$-value reported by DESeq2 | smallest rank is most significant (or smallest $p$-value) reported |
| voom_rank | rank of the $p$-value obtained by voom | smallest rank is most significant (or smallest $p$-value) reported |
| rank_sum | Sum of ranks | Combination of ranks from edger_rank, voom_rank and deseq_rank. Results are orderd by this sum, which represents the order of the most stable reported p-values |
| p_intersect | Largest $p$-value observed | This represents the intersect when a threshold is set on the p_intersect column |
| p_union | Smallest $p$-value observed | This represents the union when a threshold is set on the p_union column |
| genename | Extended gene name | e.g., alpha-L-fucosidase 2 corresponds to FUCA2 |
| symbol | Gene symbol | e.g., FUCA2 |
| kegg | kegg pathway identifier | For further analyses of pathways where annotated |
| coords | chromosomal coordinates | e.g., chr6:143494811-143511690 |
| strand | transcript strand | forward strand is +, reverse strand is - |
| width | transcript width | Reported in base pairs (bp) (transcript start to end) (e.g., 16,880 bp) |

the point weighted by fold change standard deviation, allowing assessment of deviation of fold change by different methods. In addition to providing plots before and after normalisation, if RUV is employed, plots before and after RUV correction are generated thus allowing users to assess the impact of normalisation and/or RUV. Each plotting function is accessible through the '*diag_plots*' function in consensusDE and described in a vignette that accompanies consensusDE: http://bioconductor.org/packages/consensusDE/.

## Software Comparison

For summarization of software and their features, we exclusively focus on methods that report multiple RNA-seq algorithm results in Table S1. Key criteria for software comparison were (1) ease of use, (2) features, (3) correction capability and (4) integration method.

IDEAMEX (*Jimenez-Jacinto, Sanchez-Flores & Vega-Alvarado, 2019*) and consexpression (*Costa-Silva, Domingues & Lopes, 2017*) implement a consensus-based voting approach, based on the number of algorithms reporting DE at a pre-defined significance threshold, arguing that consensus amongst multiple methods improves accuracy of DE detection. IDEAMEX, is web-based, targeted at non-bioinformaticians and requires users to click through individual steps. consexpression is an instance released for assessing results and is not readily generalizable. MultiRankSeq combines ranks derived from edgeR, DESeq and baySeq ordered p-values as an overall rank sum for reporting of results (*Guo et al., 2014*). Assessment of intersecting sets showed similar performance with DESeq and edgeR however baySeq failed to identify a similar proportion of overlapping DE genes (*Guo et al., 2014*). All software except metaseqR were developed outside of the widely used R/Bioconductor repository, requiring users to install unix-based software and in some instances install old versions of software. Rather than considering intersection of common DE genes, metaseqR implements several methods for combining p-values, (Simes, Union, Fisher's, and Whitlock weighting) in addition to a proposed weighted PANDORA method (*Moulos & Hatzis, 2015*). However, combining p-values using classical meta-analytical *p*-value methods (e.g., Fishers) performed poorly and the weighted combination of p-values (PANDORA and Whitlock) was sensitive to the selection of weights, requiring proper simulation for definition of weights. Although the weighted combination of p-values improved overall performance in some cases, it did not improve false discovery rate (FDR) using simulated data (which is often an important goal, for example, in diagnostic settings) or Area Under Curve (AUC) of real data in comparison to the intersection method (*Moulos & Hatzis, 2015*). MultiRankSeq and metaseqR also implement DEseq, rather than DESeq2. All software requires the user to specify the contrast (or comparison) and none support RUV correction.

In comparison, consensusDE, is available in the BioConductor repository, implements edgeR (*McCarthy, Chen & Smyth, 2012*; *Robinson, McCarthy & Smyth, 2010*), DESeq2 (*Love, Huber & Anders, 2014*) and limma/voom (*Ritchie et al., 2015*), which benchmark analyses find to be some of the best-performing algorithms for DE analysis (*Costa-Silva, Domingues & Lopes, 2017*; *Rapaport et al., 2013*). A key feature of consensusDE is ease of use. consensusDE does not require the user to specify a contrast of interest or specify models, instead automatically performing all possible comparisons from an annotation table provided. consensusDE also improves on existing methods by integrating RUV, which has been reported to improve sample clustering by the removal of unwanted technical variation and thereby improving biological significance (*Risso et al., 2014*). When selected, consensusDE automatically updates the underlying model after estimating unknown sources of variation from at initial GLM fit (RUVr) and including in subsequent edgeR, DESeq2 and voom analyses, consistent with its goal of "ease of use". The user has the option to flag technical replicates for merging of counts and paired samples (for paired analyses) in the annotation table. To assess the performance of consensusDE and the utility of incorporating RUV across multiple RNA-seq algorithms, we apply consensusDE with and without RUV correction to real and simulated data.

## Application of consensusDE to real RNA-seq data (with and without RUV)

RUV has been reported to stabilize fold-change and reporting of DE (*Risso et al., 2014*). To assess if RUV improved the concordance of reported DE across different algorithms, we first use the Jaccard Similarity Coefficient (JC) (Eq. (1)) to measure similarity of reported DE to the common (intersect) set of reported DE (adjusted $p \leq 0.05$). For assessment of RUV to stabilize fold change variation between multiple algorithms, we perform Bland-Altman analysis to assess bias and limits of agreement, being the mean agreement and 95% confidence intervals of logFC differences between pairs of methods, respectively. Goodness of fit of log fold-change (reported as $R^2$) and standard deviation (SD) of log fold-change in the intersecting and non-intersecting DE results are also evaluated for each algorithm. consensusDE was ran without RUV and then with RUV to incorporate the same RUV residuals into each algorithm for comparison of results. For application to real RNA-seq data, we utilize data from human "airway" smooth muscle cells comparing glucocorticoid treatment to untreated controls (*Himes et al., 2014*). This data is also available in the airway R package and used as example data in the consensusDE vignette.

Application of consensusDE to airway data identified 1,878 DE genes for voom, 2,114 for edgeR and 2,747 for DEseq2 (adjusted $p \leq 0.05$), of which 1,728 were in common (Fig. 2A). Voom shared the highest similarity to the intersect (JC = 0.92), followed by edgeR (JC = 0.82) and DEseq2 (JC = 0.63). Application of RUV increased the overall sizes of all sets reported as DE by 18% to 31% with 2,724 DE genes for voom, 2,935 for EdgeR and 3,341 for DEseq2 (adjusted $p \leq 0.05$), with the common set increasing by 31% (Fig. 2B). The Jaccard Similarity Coefficient similarity increased substantially for EdgeR (0.857 vs. 0.817) and DEseq2 (0.752 vs. 0.629), but improved only marginally for voom (0.923 vs. 0.920) (Figs. 2B–2C). Therefore RUV increased the overall set size of reported DE for each algorithm (largest for voom), but also increased the intersect and JC, thus appearing to stabilize overlap. However it must be noted that an increase in JC, without known truth, does not necessarily mean improved performance.

We next assessed agreement (or stability of fold change) between algorithms using Bland-Altman analyses, linear regression and standard deviation of fold change. Bias, being the mean shift of the differences between logFC values for each pair of algorithms and the limit of agreement (LOA), being the range of values where 95% of data laid were compared before and after RUV correction. Mean bias, with or without RUV was small (ranging from ∼0.003–0.013) and consistently decreased with RUV correction (Table S2). If RUV improved agreement, we would expect the LOAs to decrease, and indeed RUV decreased the upper and lower limits of agreement between voom, DESeq2 and edgeR, but not between DESeq2 and edgeR (Figs. 2C–2D, Table S2). DEseq2 and edgeR, however, also maintained the tightest limits of agreement irrespective of RUV. These results were consistent with correlation and standard deviation of logFC, where $R^2$ improved between DESeq2 (0.9796 vs. 0.9883) and voom, and edgeR (0.9799 vs. 0.9882) and voom, but did not increase between DESeq2 and edgeR (0.9998 vs. 0.9999) (Table S2). RUV also resulted in a more stable fold change of the DE set (Table S2). Overall fold change SD improved for each method, 0.013 (reduction of 0.004) for voom, 0.016 (reduction of 0.005) for EdgeR,
**A. Overlap (w/out RUV)**

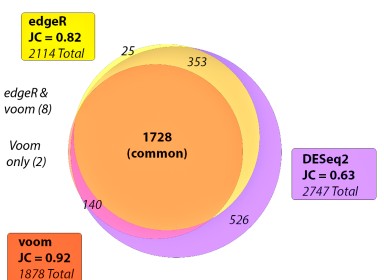

**B. Overlap (w/RUV )**

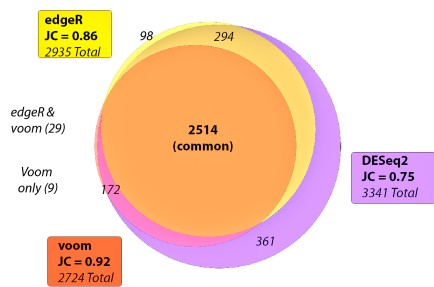

**C. Upper limit of logFC agreement**

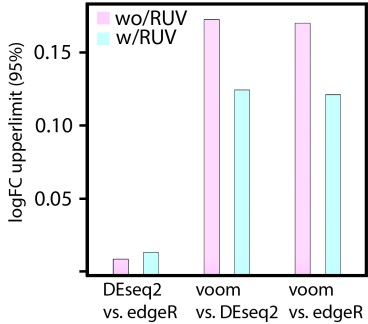

**D. Lower limit of logFC agreement**

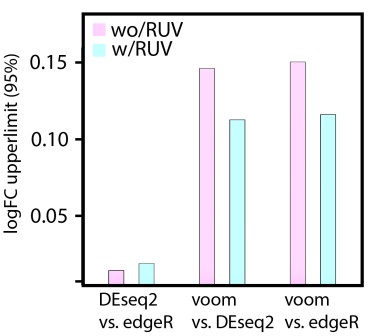

**Figure 2   Application to airway data.** (A) Jaccard Coefficient (JC) of each set to the intersect (common) without RUV correction and (B) with RUV correction. Absolute upper (C) and lower (D) limits of log fold change differences between pairs of algorithms (95% confidence intervals), with and without RUV correction.

and 0.013 (reduction of 0.003) for DEseq2 (adjusted $p \leq 0.05$). The non-intersecting set for each algorithm also improved, 0.008 (reduction of 0.004) for voom, 0.027 (reduction of 0.012) for EdgeR and 0.012 (reduction of 0.007) for DEseq2 and intersection 0.014 (reduction of 0.003). Only voom demonstrated lower fold change SD than the intersecting set compared to DEseq2 and EdgeR which exhibited higher fold change SD.

Overall, these results found that RUV (1) increased the number of reported DE genes, whilst (2) increasing the overlap of intersecting sets and (3) improving fold change agreement between methods. Thus, RUV appeared to improve the overall concordance of different methods, bringing their intersecting values closer to the union of all sets. This was consistent with a reduction of variability of fold change between algorithms, especially between voom and DEseq2 as well as edgeR. However, the smallest set of DE genes was also derived from voom and voom had the highest level of agreement to the overall intersect. Thus, it is important to note that consensus and improvement of consensus with RUV, was largely driven by voom and the additional agreement of DEseq2 and edgeR with voom after RUV application. As it is difficult to address what this means, especially JC, in the context of false discovery rate or overall performance, we next simulate data with known

numbers of DE genes and assess the utility of applying RUV in a consensus-based manner across multiple RNA-seq algorithms for improving stability of the intersecting set.

## Simulation results

For simulated data (described in Methods) we set an expected number of DE genes to 500 (or 5%), from a total of 10,000 genes with equal up/down regulation between two groups. We simulate DE with 3 and 5 replicates (in equal groups), representing commonly performed experimental designs, and report the average of 10 simulations. As per the real "airway" dataset, we begin by assessing the JC of each method, with and without RUV correction, followed by Bland-Altman analysis of bias, LOA, $R^2$ and fold change SD. Simulated data with 3 replicates (without RUV and adjusted $p \leq 0.05$) identified 304.0 DE genes for voom (JC = 0.999), 449.6 for edgeR (JC = 0.675), 438.1 for DEseq2 (JC = 0.693), of which 303.6 were in common. In contrast to real data RUV did not improve JC values, identifying 281.9 DE genes for voom (JC = 0.996), 462.2 for edgeR (JC = 0.608) and 415.5 for DEseq2 (JC = 0.676), of which 280.8 were in common (Fig. 3A and Table S2). Although JC improved overall for all methods when increasing to 5 replicates, JC was once again not improved with the application of RUV (Fig. 3A). For 5 replicates, comparing RUV vs non-RUV corrected, this identified 413.1 vs. 415.5 (JC = 0.997 vs. 0.996) DE genes for voom, 487.6 vs. 481.8 (JC = 0.845 vs. 0.859) for edgeR and 492.8 vs. 492.0 (JC = 0.837 vs. 0.841) for DEseq2 (adjusted $p \leq 0.05$), of which 413.8 vs. 411.8 were in common. Thus, replication rather than application of RUV had the greatest impact on JC in simulated data.

In contrast to the real data and albeit that the differences were negligible (a maximum mean difference of 0.008), RUV increased the mean absolute fold-change bias in all comparisons (Table S2). However, consistent with the real airway data, upper and lower limits of agreement for simulated data decreased with RUV correction for voom comparisons with DESeq2 and edgeR but not between DESeq2 and edgeR (Fig. 3B, Table S2). Notably, limits of agreement were consistently lower for simulations with 5 replicates (Fig. 3B, Table S2). Overall fold change $R^2$ for 3 and 5 replicates also improved between DESeq2 and voom, as well as edgeR and voom, but did not increase between DESeq2 and edgeR (Table S2).

The greatest gains in fold change stability for voom were found in the 3-replicate setting, doubling the improvement observed with RUV applied in a 5-replicate setting. Absolute mean fold change SD of simulated results also improved with RUV correction, 0.0086 (vs 0.011 or a reduction of 0.024) for voom, 0.0147 (vs 0.0154 or a reduction of 0.007) for edgeR, and 0.0133 (vs 0.0141 or a reduction of 0.008) for DEseq2 (adjusted $p \leq 0.05$). With the exception of edgeR, the non-intersecting set of each method and the overall intersect also improved with 5 replicates, which was 0.0534 (vs. 0.0315, an increase of 0.029) for voom, 0.0242 (vs 0.0252, a decrease of 0.01) for EdgeR and 0.0232 (vs 0.0217, an increase of 0.015) for DEseq2 and intersection (0.0085 vs 0.0107, a decrease of 0.022) (Table S2). RUV also reduced mean absolute SD with 5 replicates and the intersect had the lowest value (0.0085), but increasing to 5 replicates did not improve overall deviation of fold change, compared to 3 replicates. Thus, the greatest gains of RUV were made in a setting of 3-replicates, with diminished improvement for voom with additional replication.

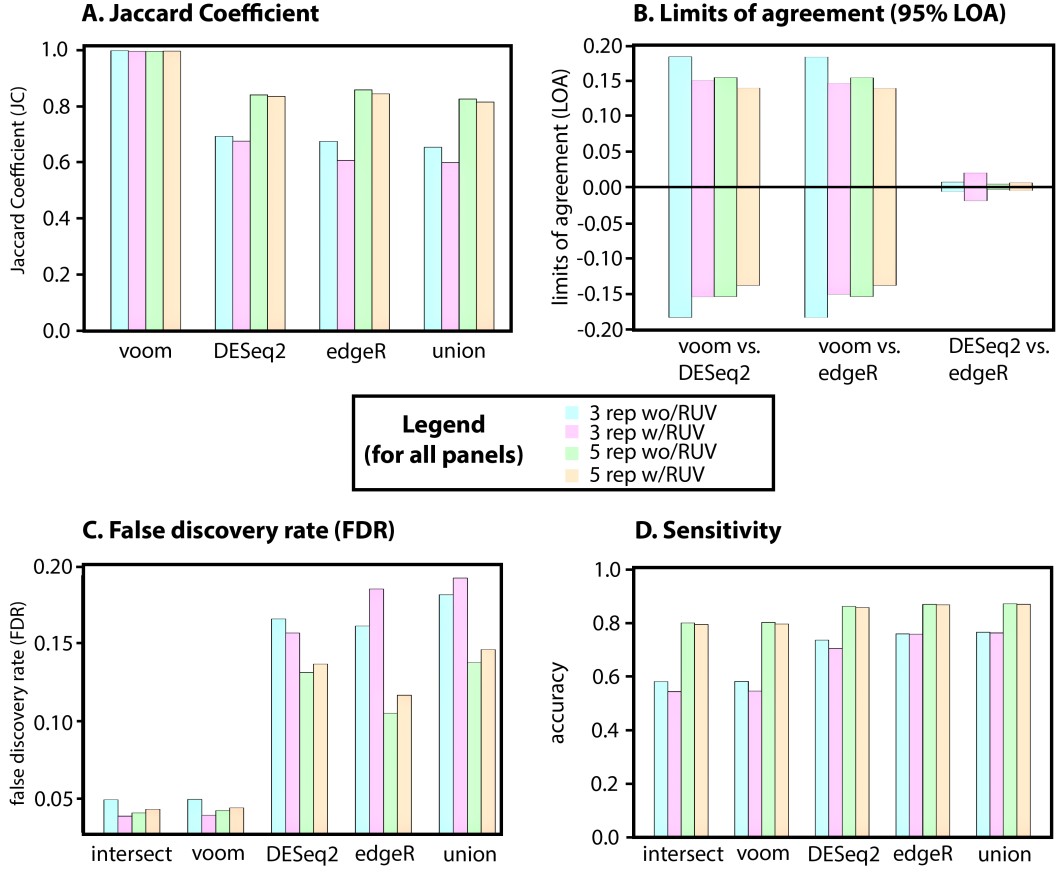

**Figure 3  Simulated data for 3 and 5 replications with and without RUV.** (A) Jaccard Coefficient (JC) of each method (B) limits of log fold change differences between pairs of algorithms (95% confidence intervals), with and without RUV correction (C) False Discovery Rates (FDR), lower number is better and (D) Sensitivity (or recall), higher number is better. Each panel contains 3 replicates, 5 replicates and without and without RUV correction–see central legend for colour coding scheme. All values represent the absolute average of 10 simulations.

As we defined a truth set (known DE genes) we assess the performance of consensusDE with and without RUV correction for each method, as well as the union and intersect. We utilise (1) False Discovery Rate (FDR) (Fig. 3C, Eqs. (4)) and (2) sensitivity (or recall) (Fig. 3D, Eq. (3)) for assessment of false positive and false negative rates. Mean FDR (of 10 simulations) was lowest for the intersect and improved with correction of RUV (0.0491 vs. 0.0384, for 3 replicates). FDR improved with the application of RUV for voom (0.0494 vs 0.0390) and DESeq2 (0.1654 vs 0.1565), but it did not improve for the union (0.1811 vs 0.1918) or edgeR (0.1610 vs 0.1847) (Fig. 3C). Although the same pattern emerged for the FDR of 5 replicates, whereby the intersect had the lowest FDR (0.0405), application of RUV did not improve the FDR for any individual method (Table S2). Although a low FDR is desirable in some instances, ensuring the number of false positives are minimised, controlling for false negatives whereby true results are incorrectly missed (or recall/sensitivity) can also be of importance. In contrast to FDR, sensitivity did not
increase with the application of RUV for any algorithm tested, indeed decreasing with RUV application and was highest in a setting of 5 replicates for the union of all methods (0.872).

Here, we establish that RUV improved FDR for the intersect for 3 replicates, approaching the same FDR as 5 replicates without application of RUV, but did not further improve FDR with 5 replicates. The improvement of FDR related to a reduction of fold-change variation and limit of agreement of different algorithms, which was greater in a setting of 3 replicates versus 5 replicates. However, the same was not true for all algorithms individually, suggesting that the greatest gain in FDR results from the combination of different algorithms. Observing the lowest FDR for the intersect amongst different algorithms and the highest for the union could be considered consistent with a model where increased stringency through inclusion of multiple forms of evidence improves FDR. Indeed, this would be consistent with previous benchmarking studies (*Costa-Silva, Domingues & Lopes, 2017*; *Guo et al., 2014*; *Moulos & Hatzis, 2015*). However, we found that voom was also the most conservative method, based on set size of DE, in both the 3 and 5 replicate experiments and that voom FDR was also the lowest. Hence voom was the main driver of the low FDR observed in the intersect. This is consistent with previous results demonstrating a lower proportion of false positive results by voom (*Ritchie et al., 2015*). Overall, RUV improved the confidence of identifying true positives, or minimizing false positives, when combined with the intersect of DE reported by multiple algorithms and experiments performed with a lower number of replicates, but with minimal gains over using voom alone. Importantly, the difference between FDR performance with the application of RUV to lower numbers of replicates (3 versus 5) diminished with increased replication. This was likely due to the improved performance of the linear model based voom without RUV correction. Although a trade-off between recall and precision was observed for voom, with voom also being the least sensitive, RUV did not improve sensitivity for any method, in 3 or 5 replicate setting. Indeed the union, being the combination (rather than intersection) of all evidence, performed better than any other individual method. This suggests that RUV is favouring improvement of FDR (or precision) through minimising false positives, but potentially at the cost of false negatives. In this context, we also note that increased JC observed is a poor metric for assessment of FDR. Real data was not replicated with our simulated data, where instead a decreased JC coincided with improved FDR. However, RUV certainly improved logFC agreement between methods, especially between voom and edgeR as well as DEseq2. Overall, these results support the application of RUV in a low-replicate setting for stabilisation of fold-change, in particular for voom and DEseq2, however RUV did not improve FDR with increased replication or sensitivity in any setting.

## CONCLUSIONS

We present consensusDE, a freely available R/Bioconductor package, that allows for simple and automated DE analysis to be performed using multiple methods, readily allowing the user to observe variability of DE due to method selection. Application of consensusDE to real and simulated data highlights the following rules, some of which are already established

in the RNA-seq community and others that require further consideration when analysing RNA-seq data. (1) the intersect of multiple methods has lowest FDR (but this can be driven by a single high performing algorithm, which we find to be voom), (2) RUV improves the intersect FDR (as well as voom and DESeq2 individually) with smaller number of replicates but this effect diminishes with increased replication, (3) RUV does not improve FDR for all individual RNA-seq algorithms, (4) RUV does not improve sensitivity, (5) the union appears to strike a balance for recovery of true positives, through minimising loss of false negatives when using multiple methods, and finally (6) increased replicate numbers, without RUV, has the best recovery of DE genes when considering FDR and sensitivity together—reinforcing increased replication for recovery of true DE genes. We do note however that this is not an exhaustive testing of all possible scenarios. For instance, it remains to be explored if these rules apply with increased modelling of noise, RUV correction methodology (here we apply RUVr), number of hidden variables or if these rules generalize to other methods for combining RNA-seq algorithms, such as weighting of $p$-values. However, our results do indicate the utility and ease of consensusDE for performing analysis with multiple RNA-seq algorithms and integration with RUV, whilst offering some rules for considering results. Future work will aim to incorporate additional algorithms and combination methods.

### Funding
This work has been supported by the NHMRC fellowship APP1139756. The funders had no role in study design, data collection and analysis, decision to publish, or preparation of the manuscript.

### Grant Disclosures
The following grant information was disclosed by the authors:
NHMRC: APP1139756.

### Competing Interests
The authors declare there are no competing interests.

### Author Contributions
- Ashley J. Waardenberg conceived and designed the experiments, performed the experiments, analyzed the data, contributed reagents/materials/analysis tools, prepared figures and/or tables, authored or reviewed drafts of the paper, approved the final draft.
- Matthew A. Field analyzed the data, prepared figures and/or tables, approved the final draft.

### Data Availability
The code is available at GitHub: https://github.com/awaardenberg/consensusDE_material.

consensusDE is available at Bioconductor and is licensed under GPL-3: http://bioconductor.org/packages/release/bioc/html/consensusDE.html.

The simulated data is available in Table S2.

## Supplemental Information

Supplemental information for this article can be found online at http://dx.doi.org/10.7717/peerj.8206#supplemental-information.

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
