# Peer review of "consensusDE: an R package for assessing consensus of multiple RNA-seq algorithms with RUV correction"

_PeerJ, doi:10.7717/peerj.8206_

## Round 0.1 · original submission · Minor Revisions

In general the manuscript was well received. The suggestions provided from the reviewers are worthwhile adopting to enhance the presentation of the manuscript. Some of the suggestions may be somewhat extreme in trying to provide a more gold standard approachable dataset; perhaps you may have run across improved datasets since this submission to appease this viewpoint. In addition, there were a few areas which appeared somewhat misleading or raising argument in its approach; it would be worthwhile trying to add clarity in some of the rough areas discussed. I will set this manuscript at a 'minor revision' status; please try to address the reviewer comments to your best ability, and address issues regarding clarity and figures where needed. It appears to be a larger request for a minor revision; however, the impact of the manuscript will be improved. Thank you for your contribution and we look forward to you revised manuscript.

Reviewer 1 ·

Basic reporting

Figure 3 is very misleading, please begin the y-axis at 0. By not beginning the y-axis of some plots at 0, bars look drastically different when they are really very similar. For example, consider Figure 3D: all bar heights are between 0.976 and 0.988, but some are only one tenth the height of another.

Experimental design

no comment

Validity of the findings

no comment

Additional comments

The paper describes a useful new package, consensusDE, which allows the user to perform differential expression analysis with 3 of the more popular methods in the literature, and compare the results. As well as providing a package to run multiple analyses with less work and easily compare the output (providing the p-values for each method is useful for users who may wish to add their own consensus method to combine the p-values), consensusDE also allows easy integration of RUV into the DE workflow.

Some specific comments:

1. One goal of the paper appears to be to demonstrate that a consensus approach improves the ability to detect DE genes. However, as the authors themselves note (lines 342-344), much of their consensus results are driven by voom, as genes called DE by voom essentially form a subset of those genes called DE by edgeR and DESeq2. Since this is the case, the results don't really show the benefits of using multiple RNA-Seq methods; just the performance of voom itself.

This does not mean that the consensus approach is a bad idea or that consensusDE is not useful (on the contrary, it is very helpful to be able to compare results easily across different methods), but that some conclusions in the paper are misleading. On lines 334-336, the authors write that "observing the lowest FDR for the intersect ... is consistent with a model where increased stringency through inclusion of multiple forms of evidence ... improves FDR". This is true, but the results are also consistent with a model where voom has the lowest FDR, and voom DE genes are the same set as the intersect, in which case including edgeR and DESeq2 don't really improve on voom. The argument for including multiple methods would be strengthened if the authors could provide a simulation in which DESeq2 or edgeR perform better than voom (see, for example, Figure 7 in Love et al.'s DESeq2 paper https://www.ncbi.nlm.nih.gov/pmc/articles/PMC4302049/).


2. Because voom DE genes are basically a subset of edgeR and DESeq2 DE genes, it is hard to interpret the Jaccard similarity coefficients. As the JC is calculated in reference to the intersect, voom naturally has a JC very close to 1 (see lines 257-258, e.g.). However, in different cases a high JC to the intersect might be good or bad. If voom is too conservative, methods with a low JC might actually be better by capturing more DE genes, whereas if edgeR/DESeq2 are too liberal the a high JC might indicate better performance by voom. Therefore, when reporting the JCs for the different methods, please explain that context is necessary to determine whether we want a high or low JC. If additional performance metrics are known (precision, recall, etc.) then we know the context, but for real data where ground truth is unknown, the relative JC values are hard to interpret.

Similarly, on lines 276-83 the authors describe how "RUV appeared to improve the overall concordance of different methods". The implication seems to be that this improvement is a good thing. However, this might not be the case. Mathematically, I can improve the JC for all methods by identifying an additional group of genes as DE in all methods. But without knowing whether this additional group is truly DE, we don't know whether actual performance has improved or not.


3. On lines 369-70, the authors write that "RUV improves the intersect FDR with smaller number of replicates but this effect diminishes with increased replication", and in the caption for Figure 3 the authors state that lower numbers are better for FDR. Please make it clear that a lower FDR is not always to be desired, if it leads to loss of power. For example, suppose we use BH to control FDR at level 0.05 for two different methods. Method 1 has an actual FDR of 0.01 and Method 2 has an actual FDR of 0.04. If both detect the same number of true positives, then we would prefer Method 1. If Method 1 detects many fewer true positives than Method 2, then we would prefer Method 2.


4. Accuracy is used to assess performance on the simulation. However, accuracy is known to be a poor metric when there are unbalanced classes, as is the case in the simulated data (5% to 95%). I recommend using pairs of precision (TP/(TP + FP) = 1 - FDR) and recall (TP/(TP + FN)) values instead.


5. In the introduction, the authors note that "there is no gold standard approach for the analysis of RNA-seq data" (line 49); this is a good point, and highlights the need for consensus-based approaches, or at least the need to compare results across methods. They later cite several papers (lines 227-230) to explain why they chose voom, edgeR, and DESeq2 as the three methods to compare in consensusDE. I believe the introduction would be strengthened by including these references in the introduction, to make the argument that while there is no gold-standard approach to DE analysis, some methods tend to perform better than others - hence we can (and should) restrict our attention to the better methods when looking for consensus.


6. There appears to be a small typo in the definition of the Jaccard similarity coefficient (Eq. 1). The JC is the ratio of the sizes of the sets, not the ratio of the sets themselves. I am also confused as to how the authors are summing the sets B, and what the sets B represent - please consider rephrasing the explanation on lines 122-4.


7. I commend the authors for making their code available and easily accessible on GitHub. This is very helpful and important for reproducibility.

Reviewer 2 ·

Basic reporting

The article is written well and the quality of the images and tables are sufficient to provide the necessary background, context and need for the software described in the text.

There are specific areas of the text that require some additional material to make the article self-contained:

1) a short description of the RUVr method
2) an exact description of the "modeled FC" (referenced in lines 178-179); does this come from a meta-analysis method? If not, how were the standard deviation calculated?

Experimental design

This is not entirely applicable. The text describes a meta-tool for the aggregation of the results of differential expression analysis using RNAseq. As a meta-tool the quality of the results critically depends on the selection of the algorithms used as targets for aggregation. 1) The major criticism in this section is why the authors chose to include only 3 algorithms for differential expression i.e. edgeR, voom and deSEQ2. I feel that the need is driven mostly by the availability of the relevant bioconductor packages (in my mind and research all 10-15 available algorithms for differential expression will fail spectacularly in specific datasets). I do ask the authors to provide a more vigorous justification (currently this is addressed in lines 227-230 of the text).

2) The analysis of the experimental airway data is not entirely convincing. The data do not include known positive and negative controls, so I do not really know what to make of the JC reported. Without a gold standard, one can assume that all algorithms are equally bad, so aggregating their results will not make anyone wiser. Do the authors have access to a dataset with positive (e.g. spiked in controls) controls that can be used? Alternative a bootstrap of the existing dataset in which the labels of the samples is permuted could be used

3) The R2 graphs are entirely inappropriate for the purpose of assessing agreement of the fold changes. The authors should switch to more appropriate methods for these analyses i.e. Blant Altman (or MA in the microarray world parlor) plots and statistics to establish the limits of agreement and bias.

Validity of the findings

The author's findings that voom is probably the best out of the methods there is well supported. The impact of replication, although highly not surprising, is welcome and enforces sound experimental design practice. The authors do not engage in wild speculation and one is comfortable that the results presented are aligned with the presentation in the text. I have no suggestion for this section, other than all the analyses and graphs reporting R2 should be redone with different methods as stated above.

---

## Round 0.2 · accepted · Accept

It appears that you have extensively looked at differential gene expression and normalization methods and have tried to make useful tools for improvement. The initial reviewers did have some critical comments which I believe you have adequately addressed. You have provided good documentation to make the software accessible and testable. I have classified this manuscript as ACCEPTED and will move it on for final approvals. I look forward to seeing this in a publication form soon, Congratulations on your efforts.